# University Sports in Moldova—A Means of Integration for Foreign Students: Challenges and Opportunities in the Context of Migration

Victoria Leșco [1], Mihail Onoi [2], Victoria Razmireț [1], Irina Volcu [1], Dan Iulian Alexe [3,*] and Ecaterina Lungu [1,4,5,*]

1 Institute of Physical Education and Sport, Moldova State University, MD-2024 Chisinau, Moldova; victorialesco18@gmail.com (V.L.); vrazmiret@gmail.com (V.R.); irinavolcu86@gmail.com (I.V.)
2 National Agency for Quality Assurance in Education and Research, MD-2028 Chisinau, Moldova; onoi.mihail@anacec.md
3 Department of Physical and Occupational Therapy, "Vasile Alecsandri" University of Bacău, 600115 Bacău, Romania
4 Department of doctoral and postdoctoral studies, "Ion Creanga" State Pedagogical University, MD-2069 Chisinau, Moldova
5 Department of Life Sciences, "Dunarea de Jos" University of Galați, 800008 Galați, Romania
* Correspondence: alexedaniulian@ub.ro (D.I.A.); lungu.ecaterina@upsc.md (E.L.)

**Abstract:** This article aims to analyze to what extent university sports can serve as an effective means of integration for foreign students and the main challenges and opportunities they encounter in the integration process in the context of migration. In this study, "integration" is defined as the process of adapting foreign students to their new academic and social environment, involving the development of interpersonal relationships and active participation in university life. "University sports" refer to the set of physical activities organized within higher education institutions, facilitating student interaction and supporting their psycho-social development. This study aims to analyze the role of university sports activities in facilitating the integration of foreign students. A quantitative analysis was conducted based on data collected through a survey applied to a sample of 134 foreign students enrolled at a university in Moldova. Among them, 25.4% were from Ukraine, 15.7% were from Romania, 17.9% were from Greece, 14.2% were from Russia, and 26.9% were from regions of the Middle East and India. Additionally, 20 interviews with foreign students were conducted to gain a more detailed understanding of their experiences regarding participation in sports competitions. Furthermore, self-assessment questionnaires on well-being were applied. Statistical results showed significant increases in the average scores on all three scales: general well-being (from 2.00 to 2.60, $p < 0.001$, d = 1.2—large effect), activity (from 1.80 to 2.10, $p < 0.05$, d = 0.75—moderate effect), and mood (from 2.10 to 2.80, $p < 0.001$, d = 1.4—large effect). These results highlight that sports activities contribute significantly not only to improving the psychological well-being of foreign students but also to their integration into the university community.

**Keywords:** university sports; integration; foreign students; migration

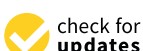

## 1. Introduction

The intensification of migration processes, especially after the outbreak of the conflict in Ukraine, has led to an increase in the number of foreign students choosing to migrate to the Republic of Moldova. They come from countries such as Ukraine (25.4%), Romania

(15.7%), Greece (17.9%), Russia (14.2%), and regions of the Middle East and India (26.9%). The significantly lower cost of living compared to Western European countries or even neighboring Romania has made it necessary to adopt a systematic and coherent approach to their integration. Data from the National Bureau of Statistics and the General Inspectorate for Migration confirm this increasing trend in temporary residence requests for study and work [1]. In Moldova, this subject has not been widely addressed in the specialized literature. There is a significant gap in the local research regarding how sports activities can facilitate the social, cultural, and academic integration of foreign students. Integration is defined as the process of adapting foreign students to their new academic and social environment, involving the development of interpersonal relationships and active participation in university life [2,3]. This study provides an opportunity to explore the impact of university sports on social cohesion and the adaptation of foreign students to the academic environment. It will bring new perspectives and strategies for effectively supporting the integration of foreign students into Moldovan universities, as these institutions represent the primary spaces for social and academic interaction for young migrants. University sports create favorable contexts for the development of interpersonal relationships and a sense of belonging.

In the international specialized literature, sports are a topic of growing interest, recognized as a potential tool for integrating and adapting foreign students. Based on interculturality theory, sports are not just a physical activity but are also a social space where individuals from different cultural backgrounds can interact, share values, and build social relationships [4]. Research conducted in Moldova [5] highlights that participation in sports activities helps foreign students overcome language and cultural barriers, thereby strengthening cohesion among student groups. This perspective is also supported by international studies [2,6,7], which argue that extracurricular activities, especially sports, play a determining role in the process of social and intercultural integration. These activities create opportunities for the development of interpersonal relationships and intercultural interaction, thereby reducing isolation and enhancing the sense of belonging to the community.

Another important aspect of this phenomenon is the link between social integration and the psychological and physical well-being of students. Participation in sports activities not only improves social relationships but also contributes to maintaining mental health, reducing the stress and anxiety associated with transitioning to a new cultural environment. Local researchers [5,8] demonstrate how sports can become an effective channel for promoting health and well-being among foreign students.

Analyzing the challenges identified in the research on foreign students' participation in university sports, the following major issues emerge: language barriers, cultural differences, limited access to resources, discrimination, racism, etc. Each of these challenges significantly impacts the integration of foreign students into university sports activities and can affect participants' cohesion in competitions and their sense of belonging to the academic community. Language barriers represent one of the most significant obstacles to the participation of foreign students in sports activities. Research [9,10] indicates that communication difficulties hinder students from effectively integrating into sports activities. They struggle not only with understanding technical instructions but also with building social relationships with their teammates. These communication barriers can reduce the sense of inclusion and create a gap between local and foreign students, affecting the performance and motivation of foreign students to actively engage in sports.

Cultural differences are another important factor affecting the participation of foreign students in sports activities. The research studies [11,12] highlight that differing cultural norms and values can cause tensions and misunderstandings. Each culture has its own norms regarding sports, collaboration, and competitiveness. When these norms are in

conflict, difficulties in communication and building trust-based relationships in various activities may arise. These conflicts can negatively impact both the integration process of foreign students into the academic community and their sense of belonging. Integration is a complex process through which foreign students adapt to the norms and values of the host society, and cultural difficulties can hinder the development of meaningful social relationships and active participation in university life. According to Tinto's theories [13], social integration is essential for students' academic and personal success, and intercultural conflicts can lead to isolation. Additionally, the sense of belonging, defined by McMillan and Chavis [14] as the feeling of acceptance and trust within a community, can be affected by these tensions. Foreign students risk feeling marginalized or excluded, which negatively influences group cohesion.

Limited access to sports resources is another major challenge, especially in small universities or in developing countries such as Moldova. Studies conducted both at the international and national levels [15,16] emphasize that limited sports resources, such as the lack of adequate equipment or sports infrastructure, can be a barrier for foreign students who need support to integrate. These limitations can lead to the lower participation of foreign students in university sports activities, reducing their opportunities for socialization and integration.

Analyzing these challenges, we can note that the participation of foreign students in university sports is affected by complex factors involving linguistic and cultural aspects as well as access to resources. To promote the effective integration of foreign students, universities should take measures to mitigate these barriers by offering language courses and cultural awareness programs and by improving their sports infrastructure. In regard to the opportunities that sports offer to foreign students, the specialized literature provides solid evidence highlighting the major benefits of engaging in sports activities, contributing to their integration and well-being [17,18].

One of the most evident advantages of participating in sports is the creation of social networks. The authors of [19,20] emphasize that sports facilitate social interactions between foreign students and their local peers, accelerating the integration process. By participating in training sessions and competitions, students not only develop their athletic skills but also could form strong relationships that provide emotional support and reinforce their sense of belonging to the community. In the university setting, this aspect is particularly important, as foreign students often face social isolation and a lack of support networks, which can lead to difficulties in adapting to academic and social life. Recent studies [21,22] show that without an adequate support network, foreign students are more exposed to the risks of dropping out and emotional difficulties. In this regard, sports can serve as a crucial platform for building these connections.

The development of intercultural competencies is another benefit highlighted in the literature [23,24]. Thus, we think that sports provide an ideal framework for students to interact with people from different cultures and learn about cultural diversity. This interaction helps develop intercultural skills such as empathy, mutual respect, and understanding of cultural differences—essential qualities in a globalized world. Sports facilitate an informal cultural exchange [11], where students can learn through direct experience, improving their ability to communicate and collaborate with individuals from diverse cultural backgrounds. In this sense, sports become an unconventional educational environment that prepares students for their future careers in multicultural societies.

Improving physical and mental health is another strong argument in favor of participating in sports activities. According to the authors Bota and Enachi [25,26], regular physical activity is essential for maintaining well-being, especially for students facing the stress of adapting to a new environment [27,28]. Cultural changes, linguistic difficulties,

and academic pressure can generate anxiety and stress for foreign students. Sports not only enhance physical health but also play a crucial role in reducing stress and anxiety, contributing to a more balanced mental state [29]. Participation in sports activities offers students a way to better manage the emotional and psychological challenges associated with transitioning to a new cultural environment.

These opportunities highlight that sports not only facilitate the integration of foreign students but also contribute to the development of essential intercultural skills and the improvement in their overall health. By promoting these activities in the university environment, educational institutions can support more effective integration and enhance the physical and mental well-being of foreign students, contributing to their academic and personal success.

In this research, we will emphasize the complexity of the interaction between sports and sociocultural integration, highlighting both the opportunities they offer and the challenges they entail, ensuring a solid theoretical framework for understanding how participation in sports activities can shape social relationships between different communities.

Furthermore, by analyzing theoretical sources on the sociocultural integration of foreign students [7,11,19], we can state that the main shortcoming of the initiatives analyzed or implemented lies in the lack of an integrated approach to the proposed activities. In this context, the necessity of including, alongside the frequently used standard methods, a significant component dedicated to physical and sports activities becomes evident. These activities significantly contribute to the integration process due to their unique characteristics, helping to reduce negative effects such as anxiety, low tolerance levels, and "culture shock", supporting the development of communication skills, strengthening self-awareness, and promoting the gradual adaptation to new living conditions in an adjusted and natural way.

The purpose of this study is to analyze the role of university sports activities in facilitating the integration of foreign students. In this context, special emphasis is placed on the challenges and opportunities that foreign students encounter in the context of migration, considering the various difficulties they may face in adapting to a new culture and academic environment. It is important to mention that the reasons students choose to migrate vary and may include economic, educational, or political factors, which also influence how they adapt to their new surroundings. For example, some students have migrated to Moldova to continue their studies in a more affordable educational system, while others are motivated by the opportunities of a more politically stable environment. These differences can generate diverse experiences and challenges in the process of cultural and academic integration and adaptation. At the same time, emphasis is placed on collaboration between universities and the local community, exploring how sports can act as a bridge between students and locals, thereby facilitating cultural and social integration.

To achieve the proposed objective, we formulated the hypothesis that participation in university sports activities significantly contributes to facilitating the sociocultural integration of foreign students. This participation not only improves their social skills but also helps build stable intercultural relationships, which are essential for successful adaptation to a new environment. This idea aligns with research demonstrating that sports can be an important catalyst for social integration, promoting collaboration, mutual understanding, and community cohesion [30,31].

## 2. Materials and Methods

This study is based on the analysis of migration legislation, statistical, and administrative data [1]. Additionally, empirical and analytical material, scientific hypotheses, and conclusions presented in researchers' works on the issues of foreign student integration were used.

*2.1. Participants*

For the selection of study participants, clear inclusion and exclusion criteria were applied. The inclusion criteria targeted foreign students enrolled at a university in Moldova who voluntarily agreed to participate and provided relevant information for the research objectives. Following the initial survey, the final number of participants was reduced to 134 due to the exclusion of some students for various reasons, including failure to meet the eligibility criteria (e.g., lack of foreign student status), refusal to participate, incomplete responses, or voluntary withdrawal from the study. The recruitment of study participants was carried out through the university's official channels, including announcements in academic groups and departments. Additionally, the recruitment process considered the number of students from each nationality to ensure sample diversity and representativeness. As a result, the final sample included 25.4% students from Ukraine, 15.7% from Romania, 17.9% from Greece, 14.2% from Russia, and 26.9% from the Middle East and India.

The interviews were conducted exclusively with male students ($n = 20$) due to their availability at the time. However, we acknowledge the importance of a gender perspective and aim to include female students in a future study to analyze differences in participation and integration in university sports and sociocultural activities.

Following the analysis of the sociological survey conducted among foreign students and the various integration processes identified, they were invited to participate in university competitions alongside their local peers, as well as in other activities organized within the institution. Additionally, university sports activities were monitored throughout a semester (January–June 2024) to observe social interactions between foreign and local students.

The activities were conducted in larger groups, as local students were also included alongside foreign students. This approach can be considered an essential condition that significantly contributes to the efficiency of the integration process. Participation in these activities took place in various settings, such as sports halls and fields, depending on the objectives set for each activity. Special attention was given to organizing dynamic and sports-related games, with the selection adapted to participants' expressed preferences.

Furthermore, an online questionnaire was administered to assess the well-being of foreign students both at the initial and final stages, aiming to analyze the impact of sports activities on their integration process.

To complete the online questionnaires, we provided instructions explaining that their participation was entirely voluntary and anonymous and that there were no right or wrong answers, as we were only interested in their perceptions of sociocultural integration and the impact of sports activities on this process. The estimated average completion time was 20 min. The research protocol was approved by the Ethics Committee of the University of Physical Education and Sport in Chișinău (Moldova, No. 01-14/22/09.01.2024).

*2.2. Variables and Measurements*

The participation of international students in university sports activities is one of the central variables referring to the degree of involvement of international students in organized sports activities within the university, including competitions, training sessions, and other similar events. Participation is measured through the frequency and type of activities that students have taken part in, which provides a clear picture of their interest and willingness to integrate through sports into the university environment. Sociocultural integration is evaluated from multiple perspectives, including the adaptation of international students to the cultural and academic context of the university, as well as the interpersonal relationships between international and local students. Additionally, the impact of sports

activities on the integration process is assessed by observing whether these activities help reduce cultural barriers and promote a more open and inclusive environment.

Online questionnaires were used to collect information about the perception of international students regarding the integration process before and after participating in sports activities. The questionnaires included questions about the impact of these activities on their well-being, cultural integration, and interpersonal relationships. Furthermore, 20 international students were interviewed to gain a deeper understanding of their experience participating in university sports activities. The data obtained from the interviews were transcribed in full and analyzed using thematic co-coding techniques. An open-coding process was applied to identify common themes, followed by axial coding to understand the relationships between these themes. The validity and coherence of the coding were ensured by verifying the inter-coding with another researcher from the team, and the discussions on coding were documented to guarantee the transparency of the process.

The statistical tool used for organizing and interpreting the data from the questionnaires was the dendrogram, which helped us identify patterns and groups of students based on their integration characteristics, thereby providing a clear picture of the diversity of international students' experiences.

To evaluate the level of integration, indicators such as the motivation for studies, language skills, satisfaction with social interactions, and access to sports activities were used. These indicators allow for a detailed measurement of various aspects of the integration process and how sports activities influence these dimensions.

The measurements of this study provide a comprehensive view of the impact of sports activities on the sociocultural integration of international students, highlighting both the benefits and challenges associated with this process.

*2.3. Statistical Analyses*

For the analysis of the responses from the questionnaire provided to international students, we used a dendrogram, which is an effective tool for organizing and interpreting data based on their similarities or distances. We used the clustering method to calculate the distance value, which, in our case, was the Euclidean distance, which is essential in the dendrogram.

SPSS 15.0 software was used to analyze the questionnaires and extract descriptive statistics about the participants, while Excel, part of Office 2019 software, was used to calculate the means, standard deviations, and correlations.

In the comparative analysis of the initial and final indices regarding the well-being of international students, we used Cohen's d coefficient to assess the effect size observed in the changes in the mean values of the three analyzed scales—well-being, physical activity, and mood.

## 3. Results

As a result of the survey conducted, we identified various integration processes that have a significant impact on the daily activities, educational process, and psycho-emotional state of the students (Figure 1, Table 1).

Figure 1 represents a dendrogram, the result of a hierarchical clustering analysis, performed based on Euclidean distance and the Single-Linkage method. On the horizontal axis, the distance values between the variables (integration indicators) are shown, illustrating how closely these were perceived by the students.

| Attributes Defining the Integration Level of Foreign Students |
|---|
| Motivation to study at a university in the Republic of Moldova |
| Specific psychological traits |
| Individual characteristics of the foreign student |
| Motivational and axiological foundation |
| Common values between the cultures in contact |
| Knowledge about the new sociocultural environment |
| Language proficiency |
| Adaptation to the new linguistic context |
| Degree of satisfaction with the quality of social and intercultural interactions |
| Existence of interactions between professors, local students, and foreign students |
| Encouragement to overcome communication barriers in the new social environment |
| Self-organization skills |
| Presence of friends who speak Romanian |
| Level of development of communication skills |
| Need for self-affirmation in a new cultural environment |
| Benefits of interactions with senior students |
| Sociocultural values of the new community |
| Understanding of behavioral norms and their specifics |
| Degree of compatibility between cultures and values |
| Possibility to socialize with people from the same country |
| Development of relevant personal and psychological traits |
| Increase in essential professional competencies |
| Access to modern learning technologies and informational resources |
| The educational environment within the university |
| Presence of a mentor or a support person |
| Organization of training and other activities for integration into the university environment |
| Regular organization of sports and recreational activities with active involvement of foreign students |
| Knowledge of the culture, history, and traditions of the host country for foreign students |
| Characteristics of life in an international community |
| Tolerance and social openness of the environment |
| Access to cultural experiences (visits to theaters, museums, exhibitions) |
| Support provided to students by staff in deans offices and departments |
| Personal development |
| Possibility to maintain a healthy lifestyle |
| Access to adequate medical services |
| Solving administrative problems, such as documentation |
| Accommodation support in dormitories |
| Support for ensuring food in cafeterias |
| Climatic–geographical conditions in the host country |
| Adequate physical preparation |
| High level of physical and mental resilience |
| Motivation to continue studies in the host country |
| Participation in scientific research activities |
| Financial stability |
| Existence of preferred recreational activities in the new environment |
| Participation of foreign students in sports, tourism, and leisure activities |
| Organizing free time for foreign students |

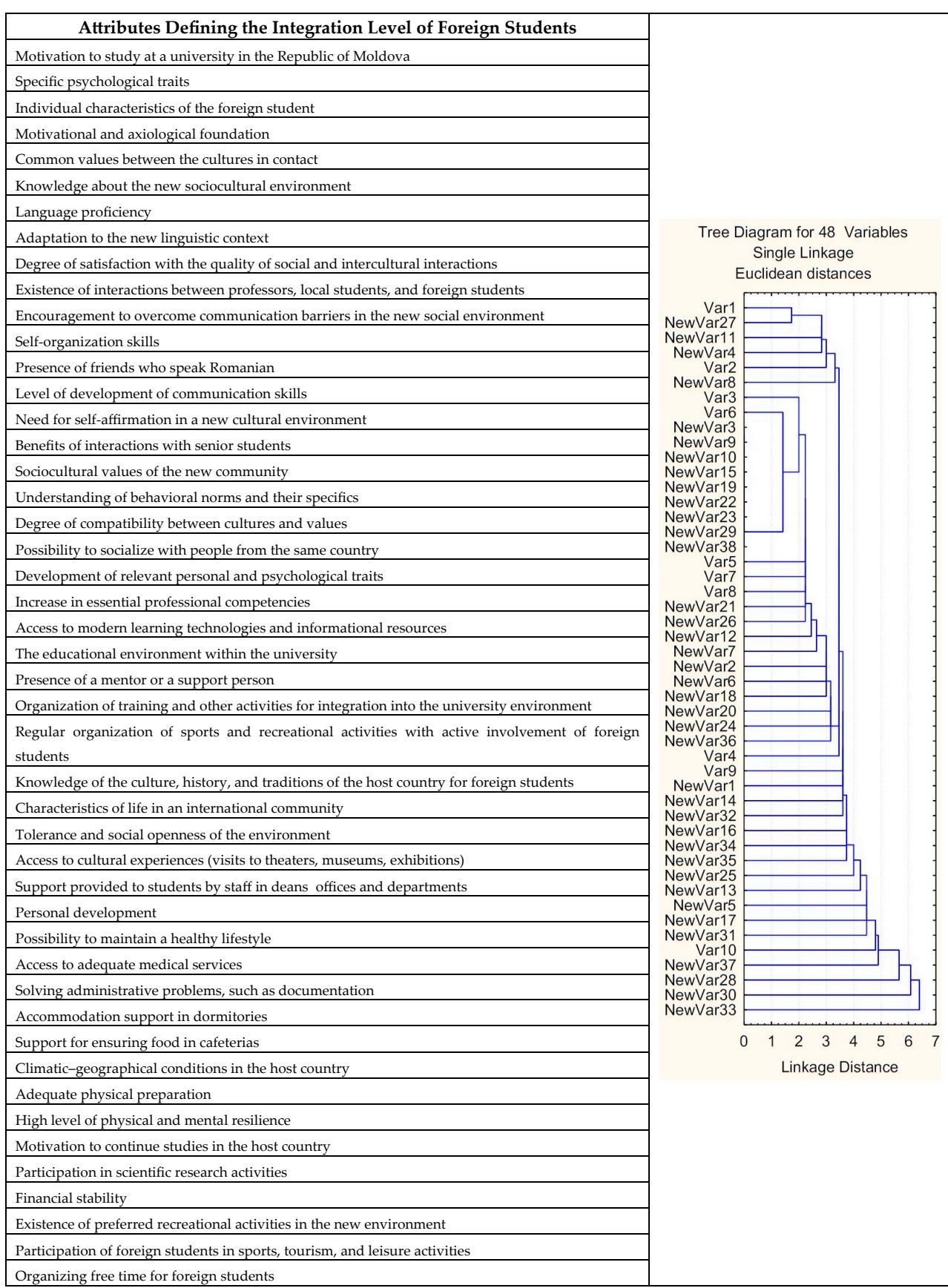

**Figure 1.** Clustering algorithms and internal and external conditions determining the indicators of foreign student integration.

**Table 1.** Impacts of factors on the main parameters of foreign students' integration (*n* = 134).

| No. | Name of Parameter | Factorial Impact Value | Factorial Impact Value (%) |
|---|---|---|---|
| 1 | Access to and ownership of living space | 0.532 | 12.57 |
| 2 | Socio-psychological factors (including the cognitive component) | 0.679 | 16.04 |
| 3 | Communication competence | 0.828 | 19.56 |
| 4 | Adaptation to the sociocultural context | 0.452 | 10.68 |
| 5 | Climatic and geographical conditions | 0.436 | 10.30 |
| 6 | Integration into the educational environment | 0.739 | 17.46 |
| 7 | Financial situation | 0.567 | 13.39 |

Thus, it is notable that the variables related to "communication and sociocultural adaptation" form a well-defined group, with a similarity distance below "1" on the linkage distance axis. These include "language proficiency", "adaptation to the new linguistic context", "frequent interactions with local students", and "communication skills developed over time". This proximity highlights the essential role that communication and integration into the social network play in the success of the adaptation process.

Another significant cluster is associated with "educational support and access to modern learning resources", defined at approximately "1.5". This includes indicators such as "presence of a mentor or support person", "access to technology and educational resources", and "support offered by academic staff". These aspects are perceived as essential for academic integration and the educational progress of foreign students. Additionally, the variables "participation in recreational, sports, and tourist activities" are grouped into a distinct cluster, formed at approximately "2". This grouping suggests that student involvement in extracurricular activities contributes to creating a sense of belonging and facilitates social integration.

Indicators reflecting "financial aspects and living conditions"—such as "financial stability", "assistance with accommodation", and "food support provided in cafeterias"—group together at about "2.5", emphasizing that economic aspects are often perceived as a common block, essential for ensuring a secure and comfortable environment that supports harmonious integration into the university setting.

In contrast, factors such as climatic–geographical conditions, physical and mental resilience, and motivation to continue studies are found at greater distances, above 4 on the dendrogram axis, indicating that these are perceived as individual influences, which are less related to the other integration indicators.

Correlating the results of this clustering analysis with the values from Table 1, we observe that the same categories of factors were identified as having high factorial impacts on the integration of foreign students. For example, communication competence has a weight of 19.56%, integration into the educational environment is evaluated at 17.46%, and the financial situation contributes with 13.39%. This overlap between the results of the clustering and factorial analyses validates the importance of these factors in the integration process.

It is important to emphasize that the integration process significantly depends on how the academic space is organized, the relevance of educational activities for international students, and the opportunities for quality interaction between them and the social and educational community. This structuring helps us understand the complex relationships between different factors and set priorities for developing a sustainable integration program through sports, tourism, and leisure activities for international students in the future.

Figure 2 and Table 2 present the determining factors and the characteristics of the integration challenges faced by international students in the university environment, according to the survey results from January 2024.

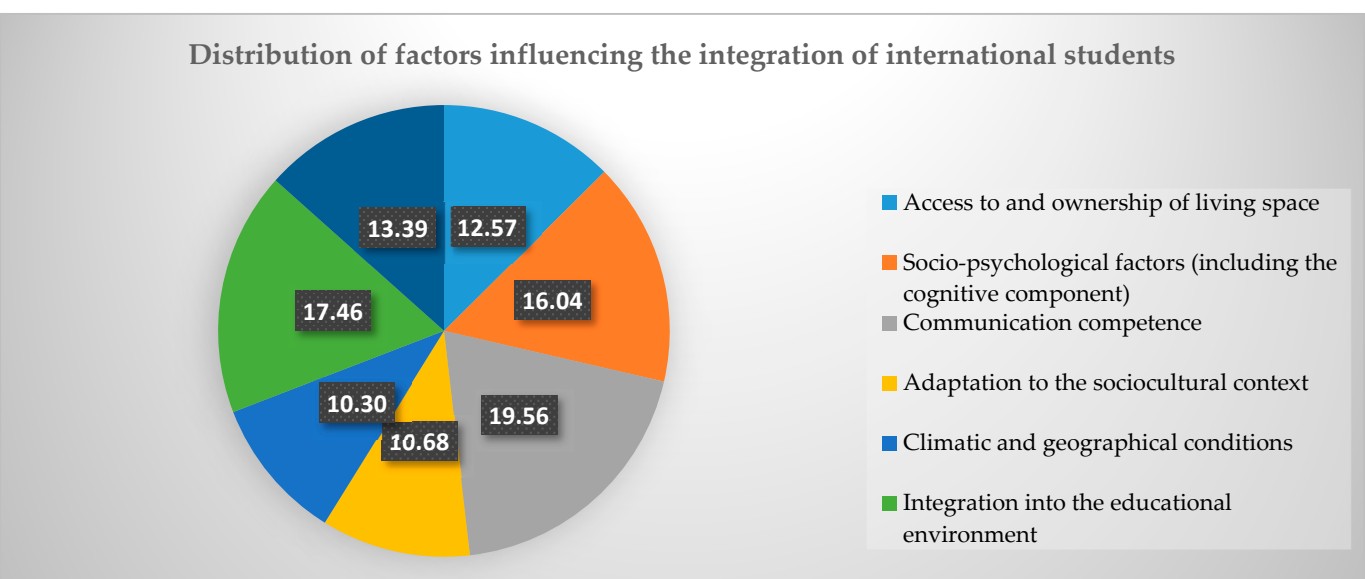

**Figure 2.** Distribution of factors influencing the integration of international students (*n* = 134).

**Table 2.** Factors and characteristics of the difficulties in the integration of international students (*n* = 134).

| Factors | Characteristics of the Difficulties |
|---|---|
| Socio-psychological factors (including the cognitive component) | Learning activities include the assimilation of information, specific requirements, material presentation, as well as the volume and complexity of tasks. Additionally, factors such as proficiency in the Romanian language, differences between the educational systems in Moldova and other countries, as well as the requirements and format of evaluations significantly influence the educational process. |
| Adaptation to the sociocultural context | Respecting specific, non-traditional norms and rules, the high level of social diversity in activities, the pronounced dynamism, sociocultural diversity, and the lack of quality recreational options. |
| Communication competence | The need to continuously interact with numerous diverse social groups, the complexity of communication with representatives of educational and administrative structures, study colleagues, and other dormitory residents, as well as daily interactions with members of society in everyday contexts—all of these take place in an urban environment with high informational density, amid a lack of communication with close ones and a feeling of isolation in a foreign country |
| Financial situation and access to a living space | The challenges of independently managing issues related to living space, budgeting, planning, and saving resources, the necessity of constant cohabitation and interaction with other students, the financial dependence on family, and the difficulty of balancing studies with a job are essential aspects of adapting to student life. |

In this context, from the perspective of organizing activities, the main mission of higher education institutions is to optimally utilize academic and sociocultural resources to create favorable conditions for the active involvement of foreign students in various types of activities. These measures aim to facilitate the process, with minimal difficulties, of one of the most complex stages of their academic formation.

The analysis of the presented factors indicates that "integration" involves the assimilation of social experience and personal development through well-structured educational activities, which support the formation of a proactive attitude in the process of learning norms, values, types of activities, social roles, and interpersonal relationships [32,33]. Thus, for complete and effective adaptation, it is essential to create conditions that encourage the active involvement of foreign students, facilitating their harmonious integration into the new social and academic environment.

Following the analysis of the sociological survey conducted among foreign students and the various integration processes identified, they were invited to participate in university competitions alongside their local peers, as well as in other activities organized within the institution. The goal was to strengthen the link between sports activities and academic ones, demonstrating that involvement in sports can contribute to a more balanced and satisfying educational experience. Also, through active participation in sports events, foreign students can feel that they are part of the university community.

Between February and April 2024, the National University Championships were held in various sports events [34]. These competitions included higher education institutions from Moldova, and the events were organized within a Physical Education and Sports Institute at a university in Moldova. The competitions encompassed a variety of sports, where students from higher education institutions competed at the national level. These events were characterized by significant intensity and moments of visual and emotional impact, generating a special reaction among the audience. Table 3 presents the calendar of the National University Championships in Moldova, offering a clear structure of the competitions and the periods allocated to each sports event.

**Table 3.** Calendar of the National University Championships in Moldova (February–April 2024).

| No. | Sports Events | Time Period |
|:---:|:---:|:---:|
| 1. | Volleyball | 26.02.2024–01.03.2024 |
| 2. | Futsal | 04.03.2024–07.03.2024 |
| 3. | Chess | 12.03.2024–14.03.2024 |
| 4. | Basketball | 21.03.2024–22.03.2024 |
| 5. | Table tennis | 26.03.2024–29.03.2024 |
| 6. | Judo | 03.04.2024 |
| 7. | Wrestling | 10.04.2024 |
| 8. | Football (soccer)–tennis | 22.04.2024–26.04.2024 |

Among the participating students, 25.4% came from Ukraine, migrating due to tensions in their country. Their involvement in the National University Championships, alongside their local peers, had a significant impact, both personally and academically. Participation in this event provided them with valuable opportunities for social integration, personal development, and improvement in their collaboration skills in a multicultural environment, thereby contributing to the enhancement of their intercultural competencies and adaptation to the university educational context.

To gain a detailed perspective on the experiences of foreign students who participated in the sports events within the National University Championships, we conducted an interview, which provided us with relevant information. Thus, we identified how well these competitions facilitated their integration into the university environment and what impact they had on their cultural and social adaptation. Table 4 visualizes data illustrating the motivations and satisfaction levels of foreign students, as well as the challenges encountered during the sports competitions.

**Table 4.** Motivation, satisfaction, and challenges of foreign students in participation in sports activities.

| No. of Students | Country | Sports Test | Motivation for Participation | Level of Satisfaction (1–5) | Challenges | Opportunities |
|---|---|---|---|---|---|---|
| S1 | Ukraine | Free fights | Integration | 4 | Competition stress | New friends |
| S2 | Kazakhstan | Kickboxing | Social adaptation | 5 | Language barriers | Improvement in abilities |
| S3 | Greece | Judo | Integration | 3 | Participation conditions | Socialization |
| S4 | Romania | Football–tennis | Health | 5 | Reduced support | Networking |
| S5 | Ukraine | Muay Thai | Physical activity | 4 | Language barriers | New knowledge |
| S6 | Russia | Badminton | Socialization | 4 | Cultural barriers | Social support |
| S7 | Ukraine | Futsal | Experience | 5 | Competition | Skills developed |
| S8 | Azerbaijan | Free fights | Cultural integration | 5 | Cultural differences | Fun academic performance |
| S9 | Ukraine | Judo | Personal development | 3 | Stress and anxiety | Integration |
| S10 | Ukraine | Volleyball | Experience | 4 | Participation conditions | Team spirit |
| S11 | Ukraine | Football | Socializing and making friends | 4 | Language barriers | Learning a new language |
| S12 | Russia | Rhythmic gymnastics | Maintaining health | 3 | Lack of time due to studies | Access to sports facilities |
| S13 | Greece | Judo | Participation in local competitions | 5 | Cultural differences in the rules of the game | Developing self-confidence |
| S14 | Romania | Wrestling | Fun and relaxation | 4 | Adaptation difficulties to a new group | Creating a healthy routine |
| S15 | Ukraine | Football | Academic support | 5 | Stress and anxiety | Networking |
| S16 | Greece | Judo | Discovering a new culture | 4 | Cultural stereotypes | Building a positive image about different cultures |
| S17 | Ukraine | Free fights | University representation | 5 | Lack of experience in organized sports | Developing a sense of belonging |
| S18 | Israel | Taekwondo | Reducing academic stress | 5 | Social isolation | Forming a social support network |
| S19 | Romania | Free fights | Professors' and dean's recommendations | 5 | Lack of previous experience | Integration |
| S20 | Ukraine | Judo | Development of sports skills | 3 | Limited access to equipment | Increasing personal performance |

Analyzing the information obtained from discussions with foreign students, we observe that their motivations to participate in the National University Championships are diverse, including reasons such as "integration" (the need to adapt to the new environment), "experience", "competition", "health", "physical activity", "socialization", and "personal development". These motivations reflect both personal desires (health, experience) and social goals (integration, adaptation, socialization). Additionally, we observe that most students rated their experience highly (scores of 4 or 5), indicating generally positive feedback. Among the main challenges encountered are "language barriers", "cultural differences", "competition pressure", "playing conditions", "limited support", etc. It is important to note that participation in sports activities brought multiple benefits to foreign students, providing them with not only opportunities for integration and personal development but also with challenges related to adapting to a new environment. The high level of satisfaction suggests that, overall, these sports experiences are formative and highly significant.

We think that sports activities, in addition to their influence on general integration processes, also contributed to improving the psychological well-being of foreign students. The psychological state was assessed through data collected from the well-being questionnaire. In Table 5 and Figure 3, the average values for the three scales (well-being, activity, mood) are shown, both at the initial and final stages.

**Table 5.** Evaluation of the well-being averages of international students during the initial and final testing (January and September 2024 ($n = 20$)).

| Scale | Initial Testing, 01.2024 | Final Testing, 09.2024 | Average Difference |
|---|---|---|---|
| Well-being | 2.00 | 2.60 | +0.60 |
| Activity | 1.80 | 2.10 | +0.30 |
| Mood | 2.10 | 2.80 | +0.70 |

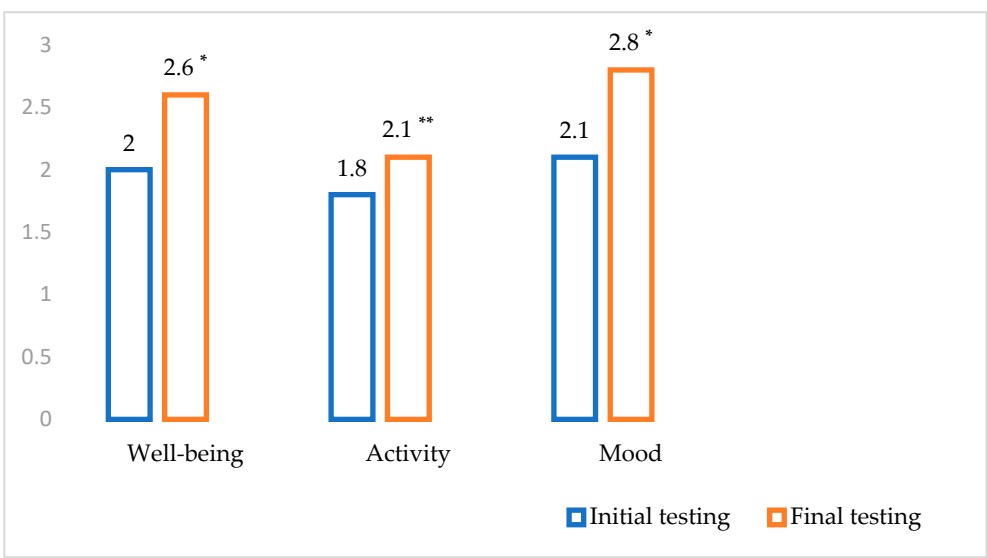

**Figure 3.** Comparative analysis of the initial and final indices regarding the well-being of international students. Note: * significant difference ($p < 0.001$) between the initial and final testing regarding well-being and mood; ** significant difference ($p < 0.05$) between the initial and final testing regarding physical activity.

Figure 3, the comparative analysis of the initial and final indices regarding the well-being of international students illustrates the evolution of the average values for the three analyzed scales—well-being, activity, and mood—during the period from January to September 2024. The data reflect a moderate increase in the mean for each of these dimensions, as follows: well-being from "2.00" to "2.60" (+0.60 points); activity from "1.80" to "2.10" (+0.30 points); mood from "2.10" to "2.80" (+0.70 points). The observed differences are statistically significant, with $p$ values $< 0.001$ for well-being and mood and $p < 0.05$ for activity, confirming an authentic progress, not a random one.

To assess the intensity of the observed changes, we calculated the effect size using Cohen's d coefficient: for well-being: d = (2.60–2.00)/SDpooled; assuming an estimated common standard deviation (SDpooled) of 0.5, the result is d = 0.60/0.5 = 1.2 (large effect); activity: d = (2.10–1.80)/SDpooled, assuming a common standard deviation of 0.4, d = 0.30/0.4 = 0.75 (moderate effect); mood: d = (2.80–2.10)/SDpooled, assuming a common standard deviation of 0.5, d = 0.70/0.5 = 1.4 (large effect). The results indicate large effects for "well-being" and "mood", and a moderate effect for "activity", highlighting that the observed changes are not only statistically significant but also practically relevant. Thus, sports activities not only improve the integration of international students but also

significantly contribute to their psychological and social well-being. This positive dynamic suggests the need to implement and strengthen sociocultural integration programs based on sports activities, tailored to the specific needs of international students. The results support the idea that constant participation in recreational, sports, and group activities facilitates stress reduction, increases intercultural cooperation, and improves overall mental and physical health. It is necessary to consider that psychological evaluations are influenced by subjective and physiological factors such as fatigue, health status, the time of day the test was taken, weather conditions, and general adaptation to the socio-academic environment.

Also, although the differences are statistically significant, the magnitude of the average increases remains moderate. The largest increase (0.7 points) is observed in "mood", indicating the better emotional well-being of the students. However, for "activity", the increase is only 0.3 points, suggesting that the level of physical engagement may require additional motivation strategies and the diversification of activity types.

In addition to quantitative data, informal observations collected during the research highlighted the following aspects: support from peers and a friendly atmosphere were essential factors in overcoming the initial feeling of isolation; communication difficulties were more pronounced among those who had no knowledge of the Romanian or English languages; participation in team sports contributed to the creation of lasting social relationships; excursions and recreational activities had a positive impact on "mood" and facilitated cultural adaptation. These findings support the idea that the social integration and emotional balance of international students are deeply influenced by the quality of their interpersonal relationships and access to sports and leisure time activities.

## 4. Discussion

The results obtained suggest that the involvement of international students in university sports activities contributes to facilitating their integration process and supports their personal development, aspects that are in line with the conclusions of previous studies on their social integration [35,36]. Participation in competitions alongside host country peers provided international students with valuable opportunities for socialization and adaptation, which are fundamental elements for an effective transition into the new educational environment.

The research identifies major challenges faced by international students, such as language barriers, cultural differences, and limited access to resources. These obstacles are consistent with studies highlighting the negative impact of communication difficulties on the integration of international students [9,37]. The literature also emphasizes cultural differences as a significant factor influencing the integration process [11,12,38]. In our study, it was found that these differences affect academic performance and can create interpersonal tensions, thus highlighting the complexity of intercultural interaction dynamics.

The quantitative analysis conducted in this study, using a sample of 134 students, showed that social factors have a significant impact on the integration process. The data highlight the importance of social aspects and communication in the integration of students into the new academic environment. The results of this analysis demonstrate that social interactions play a crucial role in maintaining the mental and physical well-being of international students.

Previous studies [37,39–41] suggest that to improve the experiences of migrants, the community should implement strategies to reduce communication barriers, cultural differences, as well as issues regarding discrimination, racism, discrepancies between the norms and values of the home country and those of the host country, and the living conditions in the host country. Our research provides a practical and dynamic contribution to the existing integration strategies, demonstrating that international students' participation in

sports activities facilitates social interaction, creates an informal environment conducive to dialogue, and promotes intercultural understanding. Through these activities, students benefit from opportunities for cooperation, empathy, and relationship building, contributing to reducing linguistic and cultural differences. In this sense, sport becomes an optimal framework for the development of human relationships and for enhancing the sense of belonging to the academic community.

The researchers D'Angelo and Makarova [4,19] emphasize the essential role of sport as a social integration factor for international students, also highlighting the challenges encountered in the context of migration. However, the data obtained in this study suggest that participation in sports activities is associated with positive experiences, characterized by a high level of satisfaction, as well as motivation from the participants. These results indicate that sport not only facilitates social integration but also contributes to improving the general perception of adaptation to new cultural and social conditions.

Moreover, our results are similar to those of the study by Chrzan-Rodak et al. [28], which highlights the importance of health and physical activity for the overall well-being of international students in Poland. Participation in physical activities not only positively influenced social integration but also contributed to their well-being.

In contrast, the study conducted by Güntaș Işık et al. [27] addresses students' attitudes toward learning a foreign language, reflecting stereotypes as obstacles in the integration process. While our study did not directly analyze stereotypes, it was found that language barriers and cultural differences are among the main challenges faced by international students, confirming the results of the literature.

It should be noted that there are several confounding factors that can influence the results regarding the social integration of international students. For example, students with significant prior sports experience may have a more positive attitude toward engaging in university sports activities, which could facilitate their faster integration. Another relevant factor is socioeconomic status; students from disadvantaged socioeconomic backgrounds may have limited financial resources for sports equipment or transportation, which could reduce their participation in sports activities. Similarly, the level of proficiency in the host country's language can directly influence students' ability to integrate into sports teams and socialize with their peers.

For the effective integration of international students, it is essential to implement sports activity programs at the local community level. These programs should focus on the active involvement of all decision-makers in promoting an inclusive educational environment that encourages the development of tolerance, understanding, and the acceptance of diversity among local community members. Thus, we suggest implementing policies and programs at the university level to support the integration of international students through sports. These include creating inclusive sports programs that combine recreational and competitive activities, adapted to the cultural diversity of the students. It is also important to organize periodic multicultural sports competitions, where teams are mixed (local and international students) to stimulate collaboration and intercultural dialogue. Additionally, introducing sports orientation sessions for international students at the beginning of the academic year would familiarize them with the university's sports infrastructure and participation opportunities. Finally, developing partnerships with local organizations to promote sports initiatives aimed at the social inclusion of young people from diverse cultural backgrounds is crucial.

At the same time, we think that, in order to improve the efficiency of sports activities in the sociocultural integration process of international students, it is necessary to develop and implement a structured sports program for them. This program should take place over

an extended period of time and include a higher frequency of physical activities, sports, and other events.

*4.1. Practical Implications*

The research findings can be used to structure sports programs that encourage interaction between foreign and local students, helping to reduce cultural barriers and facilitate intercultural communication. Similarly, higher education institutions can implement measures based on this study's conclusions to encourage the active participation of foreign students in sports, tourism, and recreational activities. Moreover, this study can serve as a support for university staff in organizing sports activities that stimulate integration and support the personal and professional development of foreign students.

*4.2. Strenghts and Limitations*

This study presents certain limitations that should be considered in future research. Although 134 participants were included, the relatively small sample size may limit the generalization of the results to the entire foreign student population. Consequently, future research aims to significantly expand the sample and focus on foreign students from all universities in the Republic of Moldova. Additionally, this study was conducted over a relatively short period (January–June 2024), but further research is planned to assess the long-term effects of sports activities on the integration process of foreign students.

This study is the first of its kind conducted in the Republic of Moldova, contributing to the development of an effective framework for integrating foreign students through sports, tourism, and recreational activities. Through this approach, we aim to deepen and optimize the existing strategies to support the integration of future foreign students who will come to study in our country.

## 5. Conclusions

Overall, this study emphasizes that foreign students' participation in sports activities not only facilitates their integration into the new sociocultural environment but also significantly contributes to improving their physical and mental well-being. These activities help increase the motivation for an active lifestyle, promoting health, preventing fatigue, and supporting intellectual performance. The results obtained from questionnaires and individual interviews highlight the positive impact of sports on strengthening social cohesion and integration, both personally and academically.

This study makes a significant contribution to the specialized literature, highlighting the importance of sports as a key factor in the integration of foreign students and proposing concrete solutions to improve their experiences at Moldovan universities. Implementing these solutions can contribute to creating a more inclusive university environment, supporting not only social integration but also the health and well-being of students.

In this context, the results of this study have significant implications for universities and policymakers. It is essential for academic institutions to adopt policies and practices that encourage the integration of foreign students through sports, thereby contributing to a more comprehensive and balanced university experience. Furthermore, future research could explore the following areas in more detail: longitudinal studies to assess the long-term impact of sports participation on the integration process of foreign students; investigating the role of different types of sports (team sports vs. individual sports) in strengthening social cohesion and the sense of belonging; examining gender differences in participation in sports activities and their effects; training academic staff to develop intercultural competencies; and exploring integration strategies in diverse cultural contexts and how they can support a faster and more sustainable integration of foreign students.

**Author Contributions:** Conceptualization, V.L. and M.O.; methodology, V.L. and D.I.A.; software, V.L.; validation, V.R., I.V. and E.L.; formal analysis, V.L. and D.I.A.; investigation, V.L.; resources, V.L. and E.L.; data curation, V.L.; writing—original draft preparation, V.L. and M.O.; writing—review and editing, V.L. and E.L.; visualization, V.L. and D.I.A.; supervision, V.L. and E.L; project administration, V.L. All authors have read and agreed to the published version of the manuscript.

**Funding:** This research has been financially supported by the Ministry of Education and Research of the Republic of Moldova and will be funded through the resources of Moldova State University (No. 060102).

**Institutional Review Board Statement:** The research protocol was approved by the Ethics Committee of the University of Physical Education and Sport in Chisinau (Moldova, No. 01-14/22/09.01.2024).

**Informed Consent Statement:** Informed consent was obtained from all subjects involved in the study.

**Data Availability Statement:** The data that support the findings of this study are available on request from the corresponding author. The data are not publicly available due to privacy or ethical restrictions.

**Conflicts of Interest:** The authors declare no conflict of interest.

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
