# Peer review of "University Sports in Moldova—A Means of Integration for Foreign Students: Challenges and Opportunities in the Context of Migration"

_societies, doi:10.3390/soc15030060_

Round 1

Reviewer 1 Report

Comments and Suggestions for Authors

This is a very interesting piece of research. It is relatively well structured, and has a hypothesis, which is dealt with by the author (s) - although I think that there is a disparity between certain sections (particularly regarding English syntax and structures used - the Introduction, for example is clearer than section 2). The following are suggestions as to how to improve on an already-good piece of research, perhaps: 

L. 21 Can the author provide backing and proof of the 'intensification of migration to Moldova'? The data from the abstract needs to be integrated, and added to here. 

L. 23-24 - There is somewhat of a jump from the notion of integration to how sports can facilitate or contribute. Perhaps the author needs to define integration first of all, then show according to existing literature how sports can fit into that. The latter part is dealt with later, but the notion of integration must be defined. 

L. 28-29 Why specifically at universities - the author should explain this?

L. 62-68 Examples of differing norms?

The notion of community belonging is dealt with but without any reference to such scholars as McMillan, Chavis, Antonisch, Colhoun, Baumeister and Leary, for example. 

Tinto , for example and student integration theories. Integration and belonging to a community are different aspects and should be outlined also. 

L. 92 Lack of a support network - proof?

L. 95 - 'we believe' - this needs to be rephrased since it is a scholarly piece and there is no question of 'belief', but proof and coherent links between what is stated from the literature and what is proved by the authors, perhaps. 

L. 106 - proof that migrants suffer from difficulty adapting to a new environment needs to be proved. 

L. 133 'foreign students' - I think there is a difference between students and their reasons for actually migrating needs to be dealt with here. 

There is a lack of interpretation or analysis of all the tables and the figures in the section dealing with the results; They are often just visually presented, and lack in 'talking the reader through the tables'. They need to be described more and detailed.  

L. 426 'we believe' - rephrase. 

Lines 440-441 repetition and reads badly. Same for L. 464.

The structure is relatively clear. But the conclusion is rather short. It needs to be developed more. It reads really simple as repetition of what has been said before. 

Comments on the Quality of English Language

On the whole, the English is relatively good, although there are discrepancies between the levels of the various sections, due to perhaps being written by multiple people. They need to clean up the syntax issues and the word order, in particular. 

Author Response

Dear Reviewer 1,

The authors thank you for your time, effort and observations. Attached, we have sent our responses, which are also included in the manuscript. With your help, we have improved the article.
Thank you

Reviewer 2 Report

Comments and Suggestions for Authors

Dear authors,
The work presented in this manuscript is of great interest to the scientific community as it uses sport as a guiding element to promote the integration of immigrant students. In this sense, sport is especially interesting to reduce communication difficulties, stress and learning difficulties.
In this study, various methodologies have been used, such as the analysis of quantitative data using interviews, which give robustness to this research.
In general, I consider this work to be very interesting.

Author Response

Dear Reviewer 2,

The authors thank you for your time, effort and observations. 

Thank you

Reviewer 3 Report

Comments and Suggestions for Authors

Review of the Manuscript: “University Sport as a Means of Integration for Foreign Students: Challenges and Opportunities in the Context of Migration”

General Comments:

The manuscript explores an important and timely topic regarding the role of university sports in the integration of foreign students. The study provides valuable insights into the challenges and opportunities associated with migration and adaptation through sports participation. The research methodology, which includes a combination of quantitative and qualitative data, strengthens the validity of the findings. However, some areas of the manuscript require further clarification, refinement, and elaboration to enhance the overall rigor and readability. Below are specific recommendations for improvement.

Specific Comments:

Title:

The title accurately reflects the study’s content, but it could be slightly more precise. For example, specifying the geographical focus (e.g., Moldova) would provide better contextual clarity for potential readers.

Abstract:

The abstract provides a clear overview of the study; however, it would benefit from greater specificity. Consider including explicit details on the study’s key findings, methodology, and the significance of the results in practical terms. Additionally, defining key terms such as ‘integration’ and ‘university sport’ in the abstract may help readers unfamiliar with the subject.

Introduction:

The introduction effectively contextualizes the research within the broader discourse on student migration and integration. However, certain aspects can be refined:

- Ensure that the research gap is explicitly stated to clarify how this study contributes to the existing literature.

- Incorporate more recent references (post-2020) to strengthen the theoretical foundation of the study.

- Reduce redundancy in defining ‘university sport’ and its role in integration.

Materials and Methods:

The methodology is well-structured and appropriate for addressing the research questions. However, the following aspects require further elaboration:

- Provide a more detailed explanation of the participant selection criteria, especially regarding how students from different nationalities were recruited.

- Clarify whether gender differences were considered in the analysis, as gender may play a role in sports participation and integration.

- Expand on how data from interviews were analyzed and coded to ensure consistency and validity in qualitative findings.

- Specify the reliability and validity measures used for survey instruments.

Results:

The results section presents relevant findings; however, the following improvements can enhance clarity and impact:

- Provide additional interpretation of statistical results, particularly linking them to real-world implications.

- Include effect sizes for significant results to provide a better understanding of the strength of observed effects.

- Consider presenting qualitative findings in a structured manner (e.g., thematic analysis) to improve readability.

Discussion:

The discussion section effectively interprets the findings but could benefit from deeper engagement with existing literature. Specific recommendations include:

- Compare findings with previous studies on sports and student integration, highlighting areas of agreement or divergence.

- Address potential confounding variables (e.g., prior sports experience, socio-economic background) that may influence integration outcomes.

- Expand on practical applications by suggesting specific policies or programs that universities can implement to enhance foreign student integration through sports.

Conclusion:

The conclusion effectively summarizes the study's key findings but should emphasize the broader implications for universities and policymakers. Additionally, it would be beneficial to outline concrete directions for future research, such as:

- Longitudinal studies to assess the long-term impact of sports participation on foreign student integration.

- Investigating the role of specific sports (e.g., team sports vs. individual sports) in shaping social cohesion.

- Exploring integration strategies in different cultural contexts.

Minor Comments:

- Ensure consistent terminology throughout the manuscript (e.g., ‘foreign students’ vs. ‘international students’).

- Review grammar and sentence structure for improved readability.

- Enhance figure and table formatting for better visual clarity.

Author Response

Dear Reviewer 3,

The authors thank you for your time, effort and observations. Attached, we have sent our responses, which are also included in the manuscript. With your help, we have improved the article.
Thank you

Round 2

Reviewer 1 Report

Comments and Suggestions for Authors

This is now much improved and the author(s) has/have addressed the elements I pointed out. There are still some careless spelling mistakes present and typos, but I suspect that MDPI will correct that once it has been accepted. 

I would say this could be published now in th present state. 

Comments on the Quality of English Language

There are some typos and spelling mistakes.